# Vocational Identity Status in Chinese Emerging Adults with and without Hearing Impairment: Latent Profiles and Relationships with Self-Esteem and Subjective Well-Being

**DOI:** 10.3390/ijerph192114473

**Published:** 2022-11-04

**Authors:** Wei Yuan, Tianxi Xu, Meimei Liu, Biying Hu

**Affiliations:** 1School of Education, Central China Normal University, Wuhan 430079, China; 2Faculty of Education, University of Macau, Macau SAR, China

**Keywords:** vocational identity, emerging adults, hearing impairment, latent profile analysis, self-esteem, subjective well-being

## Abstract

This study aimed to (1) explore the configuration of vocational identity status among emerging adults with and without hearing impairment using latent profile analysis, and (2) investigate the relationships between vocational identity status and self-esteem and subjective well-being. In total, 408 students without disabilities and 432 with hearing impairments from two Chinese higher institutions participated in the study. The Vocational Identity Status Assessment, Rosenberg Self-Esteem scale, Satisfaction with Life Scale, and Positive and Negative Affect were used to assess the major variables. The results derived five latent profiles (achieved, foreclosed, searching moratorium, undifferentiated, and diffused) of vocational identity in the present sample. The students were over-represented in undifferentiated profiles and under-represented in achieved and foreclosed ones. Hearing impairment significantly affected vocational identity status profile membership. The results showed that emerging adults with achievement and foreclosure statuses displayed healthy psychological outcomes, having the highest self-esteem, life satisfaction, and positive affect, and the lowest negative affect. In contrast, the diffused group showed the most disturbing pattern with the lowest self-esteem, life satisfaction, and positive affect, and the highest negative affect. The research findings reveal some notable issues in vocational identity status for emerging Chinese adults, raising concerns about the influence of hearing impairment on vocational identity formation, and provide implications for Chinese society to facilitate college students’ career development process to promote their vocational identity status and enhance their self-esteem and subjective well-being.

## 1. Introduction

Vocational identity formation plays important roles in adolescents’ and young adults’ academic [1,2], psychosocial [3,4,5] and career-related outcomes [6,7,8]. It is widely accepted that vocational identity refers to “the possession of a clear and stable picture of one’s goals, interests, and talents” (p. 1911) [9]. Researchers have found that individuals typically do not achieve a stable vocational identity status—the most challenging identity formation aspect for and the most critical task of emerging adults in industrialised and post-industrialised societies—during adolescence; college students, who are in emerging adulthood, continue to explore before they enter the career market [10]. Career development in adolescents and young adults with hearing impairment is under constant scrutiny. For several decades, the literature documented that adolescents with hearing impairment had lower levels of career maturity, vocational interests, and career aspiration than their hearing counterparts [11,12,13]. They encountered various career barriers, mostly because they lacked “awareness of helpful strategies or job accommodations and some had prematurely foreclosed on career choices” [12]. These studies were outdated studies mainly targeted at secondary school students; there has been little investigation of vocational identity among the incrementally larger number of students with hearing impairment receiving higher education [13].

Vocational identity has gained tremendous popularity in the last few decades, with most works examining the theoretical models, configurations, and associations of vocational identity. Existing studies across countries show discrepant results on the types and distributions of vocational identity statuses among adolescents and college students [4,14,15]. A more rigorous data analysis method, namely latent profile analysis, is required to increase the validity of vocational identity status classification results, as previous studies have exclusively used cluster analysis to categorise vocational identity. Vocational identity has attracted increasing attention in the past few years in China [16,17,18], where the profile of Chinese youth’s vocational identity status is rarely researched. The psychosocial outcomes (including self-esteem and subjective well-being) associated with vocational identity are additional research focuses. However, the relationships between vocational identity status and self-esteem and subjective well-being need to be further examined.

Based on the above background, the present study explores the configuration of vocational identity status among Chinese emerging adults with and without hearing impairment, using latent profile analysis. Then, it investigates the influence of hearing impairment on vocational identity status and the effects of vocational identity status on life outcomes, including self-esteem and subjective well-being. The results enrich the knowledge base on vocational identity status and deepen understanding of the contributions of vocational identity to adjustment in emerging adults in China.

### 1.1. Vocational Identity: Theoretical Framework and Updated Measurement

The theoretical model for vocational identity is rooted in general ego identity models. Marcia’s paradigm, a traditional identity model, suggests that identity has two dimensions—exploration and commitment [19]—that group individuals into four identity statuses: achieved, foreclosed, moratorium, and diffused. Individuals who fall into achieved status commit themselves to fully explored roles, while individuals in moratorium status are in a transitory situation that increases commitment and moves on to achievement status. However, those in foreclosed status commit to roles depending on external forces, such as parents’ will, instead of actively and independently exploring their vocational identity, while diffused status represents very low levels of exploration and commitment [20]. Contemporary models have refined and expanded Marcia’s paradigm. Luyckx and colleagues [21] divided exploration and commitment into two sets of sub-dimensions. More specifically, exploration was divided into in-breadth and in-depth exploration, and commitment into commitment making and identification with commitment. In the Crocetti model, identity formation comprises three structural dimensions: commitment, in-depth exploration, and reconsideration of commitment [22,23]. Empirical data collected based on the contemporary models captured more categories of identity statuses, in addition to the four statuses proposed by Marcia [24]. In this regard, diffused status was divided into diffused and carefree diffusion, and a new status (undifferentiated) was added based on the Luyckx model. Undifferentiated individuals are at or near the mean in all dimensions of exploration and commitment. The reconsideration dimension in Crocetti’s work contributes to the formulation of searching moratorium status, which reflects a vacillation between achieved and moratorium statuses [22].

Building on established conceptual identity status models, including Marcia’s paradigm [20], Luyckx’s model [21,25], and Crocetti’s model, Porfeli and colleagues conceptualised vocational identity as comprising six dimensions along three identity formation processes: career exploration, career commitment, and career reconsideration [26]. Each process comprises two dimensions. Career exploration involves in-breadth and in-depth exploration, referring to finding out the general characteristics of the self and the possible occupations that fit these characteristics through broad and deep exploration. Career commitment consists of making a career path decision (career commitment making) and sticking with that choice (identification with career commitment); career reconsideration involves a re-examination of previous career commitments (career self-doubt) and a comparison of available options to find one better fitting the self (commitment flexibility). 

Before Porfeli’s conceptual framework of vocational identity, Holland’s My Vocational Status was the most widely used assessment for assessing vocational identity as a unitary construct, emphasising career choice clarity and feelings instead of a stable awareness of vocational goals and interests, and overlooking the exploration process of identity formation [27,28,29,30]. Therefore, Porfeli and colleagues [2] developed the multi-dimensional Vocational Identity Status Assessment (VISA hereafter). The VISA has been validated among student samples from various countries, including the US [10,26], Italy [14,31], France [4], Korea [15], and China [18,30]. VISA’s measurement invariance has been proved across samples from different higher education settings, including universities, liberal arts colleges, and community colleges [10]; however, its applicability among students with disabilities has not been tested. 

Studies using VISA identified six vocational identity statuses—achievement, moratorium, foreclosure, searching moratorium, diffusion, and undifferentiation—among US high school and university students [10,26] and Korean university students [15]. European studies confirmed three statuses—searching moratorium, moratorium, and foreclosure—among adolescents and young adults, but yielded discrepant results regarding the classifications of achievement, undifferentiated, and diffusion [4,14,32]. Achievement cluster was absent in the Italian sample [14]; the French and Romanian studies did not identify undifferentiated status but reported two distinct, diffused statuses [4,32]. In addition to the differences in vocational identity status classification, status distribution varied in different student samples. 

Some inconsistencies might be due to social, economic, and cultural differences; for example, research has shown that nationality is a significant discriminator of identity statuses [22,33]. Therefore, it is necessary to investigate youth identity statuses in specific contexts. China has been undergoing a profound social and economic transformation in the process of industrialisation, accompanied by a rapid expansion of higher education. Li and colleagues [16] suggested that individuals’ career development trajectories in the collectivistic Chinese culture are influenced by other people’s preferences and choices rather than being independent, and vocational identity plays a vital role in this intricate process. However, the profile of Chinese young adults’ vocational identity status has not been analysed under Porfeli’s framework. 

### 1.2. Vocational Identity of Individuals with Disabilities 

The influence of disability on identity development has long been discussed [34,35]. It is argued that disability is identity constitutive and can impact personal identity [36]. Existing research on the identity of people with disabilities mainly focuses on disability identity [37,38,39], and the domain-specific vocational identity of individuals with disabilities has also attracted some academic attention. However, few studies have focused on individuals with hearing impairment. 

Forty years ago, it was reported that high school students, college students, and workers with disabilities had lower levels of vocational identity than their counterparts without disabilities [9]. A search of the updated literature only located a few studies investigating the factors influencing individuals with disabilities’ vocational identity [40,41], all of which employed Holland’s vocational identity measurement. Introducing Porfeli’s renewed and widely recognised framework to the disability research field would contribute to testing the model’s applicability in disability groups and strengthen current research on disability populations’ vocational identity. 

The lack of investigation of individuals with hearing impairment’s vocational identity leads to an incomplete understanding of the association between hearing impairment and vocational identity status during emerging adulthood. Considering the significance of vocational identity in youth career development, discussed in the next section, this study endeavours to unravel the vocational identity status of emerging adults with hearing impairment by comparing them to their hearing peers, using the updated vocational identity conceptual framework. 

### 1.3. Relationships between Vocational Identity Status with Self-Esteem and Subjective Well-Being

Over the past two decades, numerous studies have explored outcomes associated with vocational identity, with associations involving career-related outcomes [6,7,8,42,43,44,45,46] and psychosocial variables [47,48,49,50,51] gaining great popularity. Most studies were variable-centred, examining vocational identity from different process dimensions. In the last ten years, the person-centred approach has advanced the research on vocational identity by analysing how vocational identity statuses are differently related to relevant constructs. Based on relationships among vocational identity status, work valences, depression, anxiety, and distress, Porfeli and colleagues [26] categorised achieved and foreclosed as the advanced identity progress, undifferentiated and moratorium as the moderate, diffused as the delayed, and searching moratorium as the mixed. The highest scores confirmed the advanced nature of achieved and foreclosed statuses on satisfaction with life; in contrast, moratorium rather than diffusion scored the lowest on satisfaction with life [51]. 

The present study further examines vocational identity statuses’ association with life outcomes, including self-esteem and subjective well-being, given the research gaps at the domestic and international levels. At the domestic level, existing studies mainly focus on the relationships between career-related variables with vocational identity among Chinese secondary, technical college, and elite university students [6,8,16,18]. As previous Chinese studies rarely attended to the themes of psychosocial variables, this study broadens the scope of research on vocational identity in China. 

Self-esteem and subjective well-being are among the psychosocial associations of vocational identity that have attracted great attention outside China. However, the worldwide research on the associations between self-esteem and subjective well-being with vocational identity is inadequate. Self-esteem refers to individuals’ subjective evaluations of the degree to which they hold attitudes of acceptance or rejection towards themselves [52]. Proliferating studies have examined the connection between identity formation and self-esteem. Burke and Stest’s identity theory [53] lists three identity bases—social/group, role, and person; verifying different identity bases provides general senses of value, competence, and being one’s true self, and is thus linked to different self-esteem outcomes [54]. Empirical evidence supports this idea. For instance, research has shown that self-esteem is significantly predicted by social identity, personal identity, and ethnic or racial identity [55,56,57,58,59,60,61], and high self-esteem students scored significantly higher on vocational identity [62]. Nevertheless, how different vocational identity statuses associate with self-esteem remains unanswered. 

According to Diener and colleagues, subjective well-being consists of both cognitive (i.e., life satisfaction) and affective (i.e., positive affect and negative affect) components [63]. Existing studies generally support that vocational identity is significantly associated with life satisfaction, positive and negative affect [4,50,51]. However, these studies have overwhelmingly adopted Holland’s measurement of vocational identity in a variable-centred approach. It is necessary to revisit the function of vocational identity in individuals’ psychosocial adjustment using an updated, person-centred measurement framework. One study exploring the association of the cognitive component of subjective well-being (i.e., life satisfaction) with vocational identity associates diffused diffusion and moratorium statuses with lower life satisfaction levels, and achievement and foreclosure statuses with higher life satisfaction levels [4]. The affective components of subjective well-being remain to be investigated in terms of its associations with vocational identity status in a person-centred approach. Therefore, further person-centred research is necessary to deepen understanding of the psychological outcomes of different vocational identity statuses. 

### 1.4. Research Objectives 

This study’s objectives are threefold. The first is to identify the vocational identity statuses of emerging adults with and without hearing impairment under Porfeli’s framework, using latent profile analysis. The second objective is to examine the influence of hearing impairment on the vocational identity process and status membership. As prior studies have revealed that gender, age, socioeconomic background, and institution type may affect identity formation [10,33,64,65], sociodemographic variables such as mothers’ education attainment, gender, age, and institution are included in the model. The third objective is to investigate vocational identity status profiles’ relationships with self-esteem and subjective well-being. 

## 2. Materials and Methods

### 2.1. Participants and Procedure

A purposive sampling approach was applied in the current study, with 840 participating students recruited from two institutions in China—a technical college and a university—representing the two types of colleges admitting students with hearing impairment in the country under the Separate Examination Separate Admission system (SESA, “单考单招”). SESA is a college enrollment policy approved by the Ministry of Education in China, under which students with disabilities (primarily those with hearing or visual impairment) are admitted based on separate examinations organised by individual colleges and universities instead of the National College Entrance Examination. Since its introduction in 1987, SESA has become the primary means for students with hearing or visual impairment to enter college. Several specific majors are available for these students to apply through SESA. The main majors available for SESA students with hearing impairment include art and design, computer science, and special education. Although students with disabilities could also take the National College Entrance Examination, it is considered too competitive; SESA has a high admission rate and offers more support for students once they enter college. In 2017, 28 colleges and universities admitted students with disabilities via SESA [66] and students without disabilities through the National College Entrance Examination. Therefore, students with and without disabilities study and live on campus together, sharing educational resources and a humanistic education environment, promoting inclusion and helping integrate students with disabilities into society [67]. 

The sample in the present study consisted of 408 students without disabilities and 432 students with hearing impairment. The participants’ demographic information is shown in Table 1. The first author’s university gave ethical approval to the research. The participants were gathered and responded to the questionnaires in a large seminar room in their college. The researcher first gave oral instructions for the survey, with a sign language interpreter translating the instructions for students with hearing impairment. Written consent was obtained from each participant before the survey. All participants were informed about their right to withdraw from the study at any time without consequences. It took each participant approximately 15 min to complete the survey. 

### 2.2. Measures

#### 2.2.1. Vocational Identity

The VISA–Chinese Version [30] was used to assess vocational identity. Its 30 items assess vocational identity via three dimensions—commitment, exploration, and reconsideration. Each dimension is examined by two subscales: making a career commitment (four items) and identification of career commitment (six items) for the commitment dimension; in-breadth career exploration (five items) and in-depth career exploration (six items) for exploration; and commitment flexibility (five items) and career self-doubt (five 5 items) for reconsideration. A sample item is “I know what kind of work is best for me”. The respondents were asked to rate each item on a five-point Likert-type scale ranging from 1 (strongly disagree) to 5 (strongly agree). The Cronbach’s αs of the subscales in this study ranged from 0.71 to 0.82, with an overall Cronbach’s alpha of 0.81. The confirmatory factor analysis results supported the six-factor measurement model of VISA: x^2^(382) = 1058.945, *p* < 0.001, CFI = 0.925, TLI = 0.915, RMSEA = 0.046 [0.043, 0.049], SRMR = 0.052.

#### 2.2.2. Subjective Well-Being

The Satisfaction with Life Scale (SWLS) [68] and the Positive and Negative Affect Schedule (PANAS) [69] were used to evaluate the participants’ subjective well-being. The SWLS has five items measuring the cognitive component of subjective well-being: an example item is, “In most ways, my life is close to my ideal”. Two ten-item PANAS subscales measure the positive and negative affective components of subjective well-being, respectively. The positive affect items include interested, excited, strong, enthusiastic, proud, alert, inspired, determined, attentive, and active. The negative affect items include distressed, upset, hostile, irritable, scared, jittery, afraid, ashamed, guilty, and nervous. For both SWLS and PANAS, the respondents are measured using a five-point Likert scale ranging from 1 (not at all true of me) to 5 (exactly true of me). The Chinese version of SWLS and PANAS has shown satisfactory psychometric properties among college students [70]. The Cronbach’s αs for SWLS and the two PANAS subscales in the present study were 0.81, 0.84, and 0.88, respectively. The SWLS’s unidimensional factor structure was supported in the present sample: x^2^(4) = 14.772, *p* < 0.01, CFI = 0.993, TLI = 0.983, RMSEA = 0.057 [0.028, 0.089], SRMR = 0.018. The PANAS’s two-factor model was also supported: x^2^(166) = 561.295, *p* < 0.001, CFI = 0.932, TLI = 0.922, RMSEA = 0.053 [0.049, 0.058], SRMR = 0.046.

#### 2.2.3. Self-Esteem 

Global self-esteem was measured using the Rosenberg Self-Esteem Scale [52]. This scale contains ten items with a response scale ranging from 1 (does not apply to me at all) to 5 (applies to me very well). A sample item is, ‘‘I feel that I have a number of good qualities’’. Cronbach’s alpha was 0.86 in the present study. The data revealed a satisfying fit to the one-factor model of RSS: x^2^(29) = 182.609, *p* < 0.001, CFI = 0.957, TLI = 0.933, RMSEA = 0.079 [0.069, 0.091], SRMR = 0.058.

### 2.3. Data Analysis

Data were analysed using SPSS 21.0 and Mplus 8.0. Preliminary data cleaning, descriptive analysis, and reliability analysis were performed using SPSS 23. The following measurement invariance test, latent profile analysis (LPA), and regression mixture modelling (RMM) analysis were conducted in Mplus V.8.

VISA measurement invariance was tested using a multi-group confirmatory factor analysis (MGCFA) and alignment method, in order to make a trustworthy comparison between students with and without hearing impairment regarding their vocational identity status profiles. As mentioned above, existing person-centred studies on vocational identity have widely used the cluster analytic approach; however, the approach has been criticised for lacking statistically objective criteria in selecting cluster solutions [71]. The LPA technique is based on the probability model, which involves rigorous statistical tests for determining class numbers and makes choosing class criteria less arbitrary [72,73]. Therefore, LPA is used in the present study to explore the configuration of vocational identity statuses in Chinese emerging adults with and without hearing impairment, following Ferguson and colleagues’ guidance on LPA [74]. Missing data were handled using full-information maximum likelihood (FIML). For simplicity, the model used composite variables, specifically mean z-scores, as indicators instead of item-level data. As previous studies generally reported six classes of identity statuses through cluster analysis, six models ranging from one to six profiles were evaluated. Both statistical and theoretical aspects were considered to determine the optimal latent profile solution. Statistically, information criteria including AIC, CAIC, BIC, and SABIC, and inferential criteria including LMR and BLRT were used in combination, with lower AIC, CAIC, BIC, and SABIC values indicating a better fit. The non-significant LMR and BLRT test suggested no more significant improvement in model fit in the present model and could not reject the (k − 1) model; the optimal of latent classes was k, and significant LMR and BLRT indicated that the current model was better than the more parsimonious one [72]. 

The BCH method was used in mixture models to examine how covariates predicted latent profiles and how latent profile membership affected the distal outcome [75]. Covariates (school, gender, grade, mother education attainment, and disability) were added as predictor variables based on the retained profile model. After completing the regression, mixed modelling with predictive variables, self-esteem, life satisfaction, positive affect and negative affect were added as distal outcome variables. Differences in profiles’ mean distal outcome values represent the profiles’ effects on the distal outcome.

## 3. Results

### 3.1. Preliminary Analysis

Means, standard deviations, and correlations of the research variables are shown in Table 2. The absolute values of skewness and kurtosis of all variables were lower than 1, indicating that all variables were normally distributed. The correlations between the six factors of vocational identity and the outcome variables were all in line with theoretical expectation, except for non-significant correlations between career flexibility with life satisfaction and positive affect, and between in-breadth career exploration with negative affect. 

### 3.2. Measurement Invariance Test

MGCFA was conducted at first to test the measurement invariance of VISA across the two groups. The configural measurement invariance model showed an acceptable model fit (CFI = 0.920, TLI = 0.908, RMSEA = 0.048, SRMR = 0.056); the metric invariance model was also supported (CFI = 0.914, TLI = 0.904, RMSEA = 0.049, SRMR = 0.064), with ΔCFI, ΔTLI, ΔRMSEA, and ΔSRMR less than 0.01 compared to the configural model. However, the scalar invariance model was rejected (CFI = 0.889, TLI = 0.880, RMSEA = 0.054, SRMR = 0.070). An alignment analysis was further conducted to determine the non-invariant parameters. The estimates were achieved by the FIXED alignment model, taking the hearing group as the reference. The alignment analysis confirmed that all 30 items’ factor loadings were invariant in the two groups, indicating that metric invariance was satisfied. Partial scalar invariance was established, with three items’ intercepts being non-invariant (detailed results are shown in Appendix A). The three items with non-invariant intercepts include two items from identification of career commitment (i.e., item 5: “I have invested a lot of energy into preparing for my chosen career”, and item 10: “My career choice will permit me to have the kind of family life I wish to have”), and one item from in-breadth career exploration (i.e., item 24: “I am learning about various jobs that I might like”). According to Muthén and Asparouhov [76], 25% is the maximum acceptable amount of non-invariance parameters for trustworthy group comparison. The non-invariant parameters rate for identification of career commitment is 4/(2 × 5 × 2) = 20%, and for in-breadth career exploration is 2/(2 × 5 × 2) = 10%; both are smaller than 25%. Therefore, students with and without hearing impairment can be compared by their VISA score. 

### 3.3. Latent Profile Analysis Modelling

#### 3.3.1. Five Vocational Identity Profile Model 

The fit statistics of the six models are presented in Table 3. AIC, CAIC, BIC, and aBIC continued to decrease from the one-profile solution to the six-profile solution, but the differences became smaller and smaller. The significant *p*-value of the LMR tests indicated that the model fit continued to improve from the two-profile model to the four-profile model and from the five-profile model to the six-profile model. Significant BLRT tests suggested that the model fit continued to improve from the two- to six-profile models. However, the six-profile model was rejected because its smallest profile only had 1.4% (N = 13) of sample participants; as such, it was too small to be included [77], and the resultant six profiles were not theoretically justifiable. The non-significant *p*-value of the LMR test indicated that the model fit did not improve significantly from the four-profile model to the five-profile model. However, the five-profile model had lower AIC, CAIC, BIC and SABIC, and significant BLRT values. Its smallest profile proportion was about 5%, and the five profiles were all theoretically justifiable and conceptually meaningful. Therefore, the five-profile model was retained as the optimal solution in the present study. 

The conceptual fit of the latent profile models was examined by plotting the mean z-standardised scores of the six vocational identity dimensions for each latent profile. As can be inferred from Figure 1, students who showed low commitment and exploration and average reconsideration (20%) were likely to be classified into profile 1, which is defined as diffusion; profile 2 (5.7%) contains students with high commitment, low reconsideration, and high exploration, and is labelled as achievement; profile 3 (5.0%), representing foreclosure, contains high commitment (lower than profile 2) and exploration in depth, and low reconsideration and exploration in breadth. Profile 4 (60.7%) comprises participants who scored in the median in all six dimensions, representing undifferentiated status. Profile 5 (8.5%) consists of students with intermediate high commitment, reconsideration, and exploration, indicating that they are in the searching moratorium status.

The distribution of the five vocational identity statuses in students with and without hearing impairment is presented in Figure 2. It can be seen that the percentages of diffusion, undifferentiated, and searching moratorium were higher in students with hearing impairment, while the percentages of achievement and foreclosure were lower. 

#### 3.3.2. Effects of Covariates on Profile Membership 

The effects of covariates—including institution, age, gender, mother’s educational attainment, and hearing impairment—on latent profile membership were examined using the BCH method. The results showed that only the mother’s educational attainment and hearing impairment significantly affected vocational identity profile membership. Specifically, as shown in Table 4, the mother’s education level had a significant positive effect on the odds of being in the achievement (OR = exp(0.40) = 1.49, *p* < 0.01) and searching moratorium (OR = exp(0.32) = 1.38, *p* < 0.05) than in the undifferentiated one, meaning students whose mother held a higher educational degree were more likely to be in the searching moratorium and achievement profiles than in the undifferentiated profile. In addition, hearing impairment had a significant positive effect on the odds of being in the diffusion (OR = exp(0.58) = 1.79, *p* < 0.001), undifferentiated (OR = exp(0.56) =1.75, *p* < 0.001), and searching moratorium (OR = exp(0.66) = 1.93, *p* < 0.001) profiles than in the achievement profile. No other significant effects were found.

#### 3.3.3. Effects of Profile Membership on Distal Outcomes

The effects of vocational identity profile membership on distal outcomes, including self-esteem, positive affect, negative affect, and life satisfaction, were examined using the BCH method. The results (see Table 5) showed that vocational identity profile membership significantly affected all four distal outcomes. Specifically, achievement and foreclosure scored highest on self-esteem, followed by the searching moratorium, undifferentiated, and diffusion profiles, successively. Achievement, foreclosure, and searching moratorium scored the highest in life satisfaction and positive affect, diffusion scored lowest, and undifferentiated scored in between. Regarding negative affect, diffusion scored highest, achievement and foreclosure scored lowest, and searching moratorium and undifferentiated scored in between.

## 4. Discussion

The present study examined (1) vocational identity profile among a sample of Chinese emerging-adult college students with and without hearing impairment, (2) the influence of hearing impairment on vocational identity status, and (3) the effects of vocational identity status profiles on self-esteem and subjective well-being. The results derived five latent profiles (achieved, foreclosed, searching moratorium, undifferentiated, and diffused) of vocational identity in the present sample. The students were over-represented in undifferentiated profiles and under-represented in the achieved and foreclosed ones. Hearing impairment significantly affected vocational identity status profile membership. The results showed that emerging adults with achievement and foreclosure statuses displayed healthy psychological outcomes, having the highest self-esteem, life satisfaction, and positive affect, and the lowest negative affect. In contrast, the diffused group showed the most disturbing pattern, with the lowest self-esteem, life satisfaction, and positive affect, and the highest negative affect. 

Using the six vocational identity factors, the present study derived five vocational identity profiles in emerging adults with and without hearing impairment. While the profiles were generally similar to the clusters resolved in previous studies conducted among North American [10,26], European [4,14], and Korean [15] college samples, the findings differ in three important respects. First, the present study did not find a moratorium status. Second, undifferentiated status accounts for nearly two-thirds of the current sample, which is about two to three times higher than in the prior literature. Third, Chinese emerging adults are far more under-represented in achievement and foreclosure status than the aforementioned college samples elsewhere. There are several potential reasons for the disparities between the vocational identity in the present sample and the prior literature. One concerns the time of the study. Unlike previous person-centred studies, the present survey was conducted during the COVID-19 pandemic, which may have adversely affected young adults’ vocational identity formation, primarily due to reduced internship opportunities and limited social interactions. Therefore, many students were in an undifferentiated position, waiting to explore their career interests further before committing. The absent moratorium status in the present sample might have merged into the relatively large percentage of undifferentiated status students. 

The low percentage of foreclosed students in the present sample contrasts the viewpoint that foreclosure status is common in a collectivist society [24,78], as individuals’ career choices are usually influenced by normative social pressures [16,24]. A likely inference is that young adults’ values change during cultural and social transformations. It is argued that due to China’s previous one-child policy, economic reform and opening-up policy, as well as the rapid development and popularisation of the Internet, contemporary Chinese college students’ social mentality has shown a pluralistic trend with decreasing collectivism values [79,80]. Chinese young people born in the 2000s tend to be more individualistic, care more about self-fulfilment, and are more inclined to express themselves actively and stick to their ideas [81]. Unlike previous generations, they are more rebellious and less likely to commit to roles based on external forces. 

This generation, called the Buddhist-style generation, features low levels of desire [82]. Recently, a kind of “lying flat” (tang ping, “躺平”) mentality with the value orientation of not thinking ahead, muddling along, decadently giving up, and remaining motionless has been spreading among Chinese young people [83]. Chinese scholars believe that "lying flat" reflects young people’s feelings of failure and frustration due to involution (neijuan, “内卷”) [82]. The present participants, recruited from relatively low-ranking colleges in China, would suffer more extensive frustration from involution and be more likely to choose to “lie flat”. Thus, it is natural that they lack the motivation to actively explore career interests, think about career choices, and make career commitments. This might explain their low presence in achieved and foreclosed status profiles and high presence in undifferentiated. 

The results highlight the significant negative influence associated with hearing impairment in light of vocational identity. Specifically, the results indicate that students with hearing impairment were more likely to fall into diffusion, undifferentiated, and searching moratorium statuses instead of achievement, echoing the view that adolescents with hearing impairment are less mature in career development than their hearing counterparts [11,84,85]. Students with hearing impairment often reconsider their job or career choices during college and are less likely to be committed to a career than their hearing counterparts. Students with hearing impairment are severely under-represented in achieved and foreclosed status compared to students without hearing impairment in the present sample and existing studies outside China [4,10,14,15]. Hearing impairment seemed to have hindered the vocational identity development process. This is plausible because students with hearing impairment may have fewer opportunities to fully explore their career interests due to physiological and social factors. Only a few majors are available to them in colleges in China [13], and the narrow range of expected occupations upon graduation could adversely affect their career decisions. Since no research has yet compared the vocational identity profiles of youth with and without hearing impairment, the present study’s findings are only exploratory, and its explanations are only post hoc. 

The five identity statuses were externally validated by examining their associations with self-esteem and subjective well-being. The results showed that emerging adults in the achievement and foreclosure statuses displayed positive psychological outcomes, having the highest self-esteem, life satisfaction, and positive affect, and the lowest negative affect. In contrast, the diffused group demonstrated the most disturbing pattern, with the lowest self-esteem, life satisfaction, and positive affect, and the highest negative affect. This agrees with prior studies’ findings that adolescents in achievement and foreclosure identity statuses reported equally few psychosocial problems, while high well-being and diffusion status showed the least favourable psychosocial and well-being outcomes [4,22,26,86]. On the one hand, the emerging adults with searching moratorium status reported a high level of life satisfaction and positive affect, which was similar to achieved and foreclosed individuals. On the other hand, they scored significantly higher on negative affect and lower on self-esteem. This group also scored similarly to undifferentiated status students on negative affect but exhibited better self-esteem, life satisfaction, and positive affect. These findings support the view that searching moratorium is a mixed-identity progress [26]. College students’ reconsideration of career commitment, which is characterised as flexibility and doubt, may lead to favourable and unfavourable outcomes [26]. 

## 5. Conclusions

This study significantly contributes to the existing research in the following four aspects. First, it initiates efforts to explore the vocational identity of Chinese college students using a person-centred approach, deriving a five-profile solution for vocational identity statuses. This is the first study to examine identity statuses based on VISA using LPA, as extant studies classifying college students’ vocational identity adopted cluster analysis. By using LPA, this study provides more robust empirical evidence for classifying vocational identity statuses. The disparities between the present study and prior studies in other countries raise concerns about the influence of data analysis techniques on the results and cultural differences in emerging adults’ vocational identity statuses. Second, the study examined the influence of hearing impairment on vocational identity, adding new knowledge for understanding the difference in vocational identity formation between students with hearing impairment and hearing students in the Chinese context. Third, as the first study to validate VISA among students with hearing impairment, its measurement invariance tests showed evidence of VISA being interpreted similarly across students with and without hearing impairment in China. Fourth, the results regarding vocational identity status’ associations with self-esteem and subjective well-being confirmed that youths with achievement and foreclosure profiles are at the advanced statuses, those with undifferentiated profiles are in the moderate zone, those with diffused profiles are in undesirable conditions, and those with searching moratorium profiles are in a mixed state.

The current study reveals some notable issues in vocational identity status for Chinese emerging adults. However, limitations must be acknowledged. Firstly, this is a cross-sectional study, and the findings are only exploratory. Vocational identity status’ associations with hearing impairment, self-esteem, and subjective well-being are relational rather than causal. Moreover, the potential sample bias inherent in purposive sampling may limit the generalisability of the results. For balance, the present study recruited participants without hearing impairment who were attending the same colleges as students with hearing impairment. Colleges admitting students with hearing impairment in China through the SESA policy are relatively low-ranking in China, and their students are often more disadvantaged in future job markets than those from elite universities. Thus, they may not represent all Chinese higher education students, especially those in more prestigious universities. 

Despite its limitations, this study has important implications for future practice and research. Specifically, the Chinese government, colleges, and individuals with hearing impairment are encouraged to make changes to address the challenges that college students usually encounter in advancing vocational identity, self-esteem and subjective well-being. For example, more favourable policies should be developed to facilitate college students’ career development, and thus, promote their vocational identity status and enhance their self-esteem and subjective well-being. In addition, college career-counselling services are urged to develop effective programs to improve students’ vocational identity by providing more career exploration workshops and opportunities to facilitate students with hearing impairment’s vocational identity construction. These services should take young people’s changing values into account. It is even more urgent that special attention be paid to college students with hearing impairment’s vocational identity development. Moreover, students with hearing impairment are encouraged to take active initiatives to broaden their career development paths and obtain more autonomy in choosing majors, which are of crucial importance for their career development and psychological health. Future research, such as longitudinal experimental studies, are recommended to explore the factors influencing students with hearing impairment’s vocational identity development and identify effective interventions.

## Figures and Tables

**Figure 1 ijerph-19-14473-f001:**
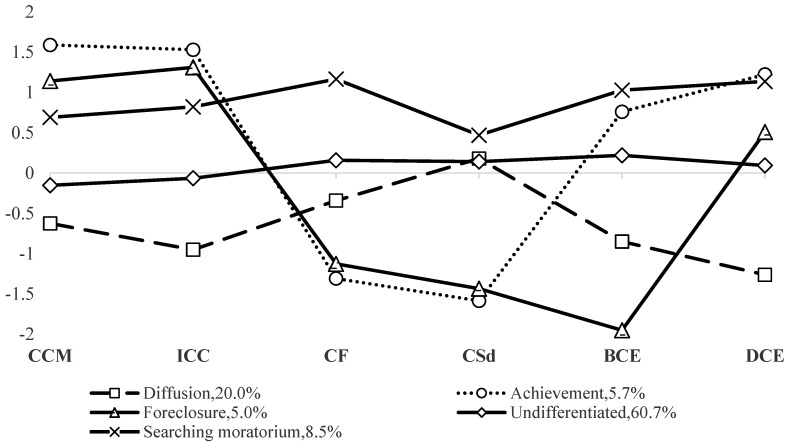
The five profiles characterised by their patterns (mean z-scores are shown). Note: CCM = career commitment making; ICC = identification with career commitment; CF = commitment flexibility; CSd = career self-doubt; BCE = in-breadth career exploration; DCE = in-depth career exploration.

**Figure 2 ijerph-19-14473-f002:**
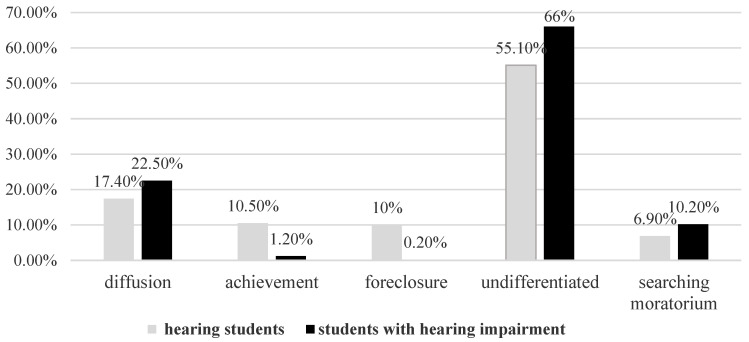
Percentage of students in the five identity statuses, separated by hearing impairment.

**Table 1 ijerph-19-14473-t001:** Demographic information.

Group		Hearing Students	Students with Hearing Impairment
Age Range (M, SD)		18~22 (19.06, 0.94)	18~25 (21.19, 1.75)
Gender	Female	364	229
	Male	39	203
	Missing	5	0
Institution	Technical college	173	177
	University	235	254
	Missing	0	1
Mother education attainment	Junior high school or below	258	352
	Senior high school or above	148	67
	Missing	2	8
Family monthly income	Below 5000	148	281
	5000~10,000	172	117
	Above 10,000	68	27
	Missing	20	7
Total Number		408	432

**Table 2 ijerph-19-14473-t002:** Descriptive statistics of the key variables.

Variable	M	SD	1	2	3	4	5	6	7	8	9
1 CCM	3.43	0.69									
2 ICC	3.69	0.64	0.55 **								
3 CF	3.45	0.67	−0.13 **	−0.04							
4 CSd	2.91	0.87	−0.26 **	−0.35 **	0.40 **						
5 BCE	3.47	0.69	0.11 **	0.17 **	0.28 **	0.16 **					
6 DCE	3.80	0.60	0.38 **	0.53 **	0.13 **	−0.17 **	0.46 **				
7 SE	3.71	0.59	0.36 **	0.45 **	−0.12 **	−0.46 **	0.10 **	0.43 **			
8 LS	3.63	0.77	0.22 **	0.37 **	−0.04	−0.25 **	0.08 *	0.23 **	0.43 **		
9 PA	3.51	0.61	0.28 **	0.42 **	−0.06	−0.31 **	0.18 **	0.39 **	0.59 **	0.56 **	
10 NA	2.59	0.78	−0.17 **	−0.21 **	0.19 **	0.48 **	0.03	−0.19 **	−0.45 **	−0.24 **	−0.21 **

Note: CCM = career commitment making; ICC = identification with career commitment; CF = commitment flexibility; CSd = career self-doubt; BCE = in-breadth career exploration; DCE = in-depth career exploration. SE = self-esteem; LS = life satisfaction; PA = positive affect; NA = negative affect. * *p* < 0.05. ** *p* < 0.01.

**Table 3 ijerph-19-14473-t003:** Latent profile analysis fit indices.

	AIC	CAIC	BIC	SABIC	Entropy	LMR	BLRT
1 profile	14,320.90	14,389.7	14,377.70	14,339.59	Na	Na	Na
2 profiles	13,742.65	13,851.58	13,832.58	13,772.25	0.90	<0.001	<0.001
3 profiles	13,351.71	13,500.78	13,474.78	13,392.22	0.79	<0.05	<0.001
4 profiles	13,193.85	13,383.05	13,350.05	13,245.26	0.82	<0.05	<0.001
5 profiles	13,058.73	13,288.07	13,248.07	13,121.04	0.84	>0.05	<0.001
6 profiles	12,978.67	13,248.14	13,201.14	13,051.88	0.85	<0.01	<0.001

Note: Entropy is not weighted/valued in the profile enumeration process.

**Table 4 ijerph-19-14473-t004:** Effects of covariates on latent profile membership.

Undifferentiated as the Reference Group				
Variable	Diffusion	Achievement	Foreclosure	Searching Moratorium
Mother education attainment	--	0.40 **	--	0.32 *
Achievement as the reference group				
Variable	Diffusion	Undifferentiated	Foreclosure	Searching moratorium
Hearing impairment	0.58 ***	0.56 ***	--	0.66 ***

Note. * *p* < 0.05; ** *p* < 0.01; *** *p* < 0.001. -- indicates non-significant estimates.

**Table 5 ijerph-19-14473-t005:** Multiple group analysis of the differences between latent profiles in self-esteem and subjective well-being using the BCH method.

	Diffused (1)	Achieved (2)	Foreclosed (3)	Undifferentiated (4)	Searching Moratorium (5)	Differences between Profiles	x^2^(p)
	M (SE)	M (SE)	M (SE)	M (SE)	M (SE)		
Self-esteem	3.30 (0.05)	4.55 (0.08)	4.39 (0.09)	3.66 (0.02)	4.04 (0.08)	2 = 3 > 5 > 4 > 1	284.26 (<0.001)
Life satisfaction	3.23 (0.06)	4.12 (0.12)	4.13 (0.14)	3.64 (0.04)	3.85 (0.13)	2 = 3 = 5 > 4 > 1	81.20 (<0.001)
Positive affect	3.09 (0.05)	4.2 (0.10)	3.93 (0.13)	3.50 (0.03)	3.85 (0.10)	2 = 3 = 5 > 4 > 1	155.17 (<0.001)
Negative affect	2.83 (0.06)	1.9 (0.12)	1.84 (0.15)	2.63 (0.04)	2.67 (0.12)	2 = 3 < 5 = 4 < 1	82.10 (<0.001)

## Data Availability

Data are available on request due to restrictions, e.g., privacy or ethical. The data presented in this study are available on request from the corresponding author. The data are not publicly available due to the privacy of the people who took part in this research.

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
