# Peer review of "Vocational Identity Status in Chinese Emerging Adults with and without Hearing Impairment: Latent Profiles and Relationships with Self-Esteem and Subjective Well-Being"

_ijerph, 2022, doi:10.3390/ijerph192114473_

Round 1

Reviewer 1 Report

The authors have tackled a very topical subject, not only during the Covid epidemic, but in every time. The literature is rich and also based on recent researches. A summary table would have been effective in presenting each theory.

The description of the statistical methodology is up to requirements and the conclusions are easy to follow. The work meets publication standards.

Author Response

We appreciate the positive feedback and value the advice made.

We have modified descriptions of the theories mentioned in the literature review part to make them effective.

In addition, the manuscript has been proof-read to increase accuracy and clarity.  

Many thanks.

Reviewer 2 Report

The manuscript explores vocational identity status in Chinese emerging adults with and without hearing impairment, and discusses relationships between vocational identity status with self-esteem and subjective well-being. The research has high value, and thanks to the authors 'efforts. Some of my concerns about the study are as follows.

1. In the abstract section, please briefly describe the value and application of the research findings.

 2. In the introduction part, it is recommended to define the two concepts of self-esteem and subjective well-being, and supplement the reasons for selecting these two variables.

 3. ”Relationships between vocational identity status with self-esteem and subjective well-being” is an important part of the whole study, but the review of relevant studies on self-esteem and subjective well-being are still Insufficient. Whether life satisfaction, positive and negative affect can be as part of subjective well-being? Is there relevant research support? In addition, it is recommended to add some researches in recent 5 years in the literature section.

 4. In the measurement part, only Subjective well-being is mentioned to measure with Likert 5 points scale, what about the measurement of the rest variables? Positive and Negative Affect Schedule (PANAS)?  Please give specific item example.

 5. Authors are suggested to report confirmatory factor analyses first, followed by correlative analyses.

 6. The authors have given a detailed result explanation in the conclusion section, but  it is suggested to add some discussions on how these findings can be applied to individuals, organizations, or societies.

 7. The main variables in the study were measured by self-rating scales, whether the author has considered how to reduce the influence of homology bias.

Author Response

Please see the attachment for point-by-point response to the reviewer's comments. We are mostly grateful for the advice and comments received. 

Reviewer 3 Report

Overall the paper makes a contribution. Perhaps to improve the submission more literature could be added to develop the depth of the review with some further discussion around the sample selection. Otherwise I am happy with the level of rigour and discussion.

Author Response

Thank you very much for your kind words. We have incorporated more literature and tried to clarify more about the sample selection in the revised version of the manuscript.  

Round 2

Reviewer 2 Report

The authors have made extensive revisions and the manuscript has been greatly improved.

Author Response

Many thanks for your kind words re the improvement we made in the revision.